# BiMaCoSR: Binary One-Step Diffusion Model Leveraging Flexible Matrix Compression for Real Super-Resolution

Kai Liu [* 1]   Kaicheng Yang [* 1]   Zheng Chen [1]   Zhiteng Li [1]
Yong Guo [2]   Wenbo Li [3]   Linghe Kong [† 1]   Yulun Zhang [† 1]

## Abstract

While super-resolution (SR) methods based on diffusion models (DM) have demonstrated inspiring performance, their deployment is impeded due to the heavy request of memory and computation. Recent researchers apply two kinds of methods to compress or fasten the DM. One is to compress the DM into 1-bit, aka binarization, alleviating the storage and computation pressure. The other distills the multi-step DM into only one step, significantly speeding up the inference process. Nonetheless, it remains impossible to deploy DM to resource-limited edge devices. To address this problem, we propose **BiMaCoSR**, which combines binarization and one-step distillation to obtain extreme compression and acceleration. To prevent the catastrophic collapse of the model caused by binarization, we propose **s**parse **m**atrix **b**ranch (**SMB**) and **l**ow **r**ank **m**atrix **b**ranch (**LRMB**). Both auxiliary branches pass the full-precision (FP) information but in different ways. SMB absorbs the extreme values and its output is high rank, carrying abundant FP information. Whereas, the design of LRMB is inspired by LoRA and is initialized with the top $r$ SVD components, outputting low rank representation. The computation and storage overhead of our proposed branches can be safely ignored. Comprehensive comparison experiments are conducted to exhibit BiMaCoSR outperforms current state-of-the-art binarization methods and gains competitive performance compared with FP one-step model. Moreover, we achieve excellent compression and acceleration. BiMaCoSR achieves a $23.8\times$ compression ratio and a $27.4\times$ speedup ratio compared to the FP counterpart. Our code and model are available at https://github.com/Kai-Liu001/BiMaCoSR.

[*]Equal contribution [1]Shanghai Jiao Tong University [2]Huawei Consumer Business Group [3]The Chinese University of Hong Kong. Correspondence to: Linghe Kong <linghe.kong@sjtu.edu.cn>, Yulun Zhang <yulun100@gmail.com>.

*Proceedings of the $42^{nd}$ International Conference on Machine Learning*, Vancouver, Canada. PMLR 267, 2025. Copyright 2025 by the author(s).

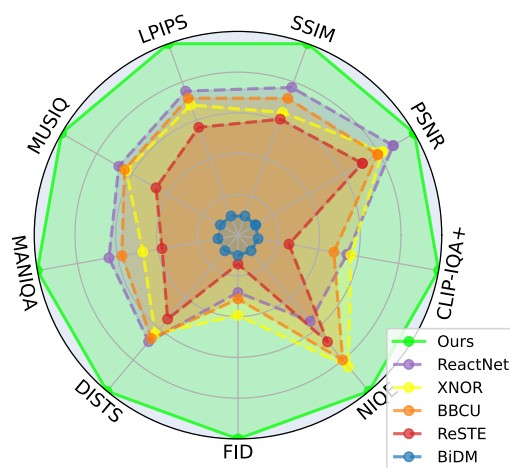

Figure 1: Performance comparison between binarization methods on the RealSR dataset. BiMaCoSR achieves consistently leading scores on all evaluation metrics.

## 1. Introduction

Single image super-resolution (SR) (Keys, 1981; Zibetti & Mayer, 2007; Lu et al., 2013; Dong et al., 2014; Yang et al., 2014; Dong et al., 2016) is a traditional yet challenging low-level vision problem. Serving as a fundamental research task, it has attracted long-standing and considerable attention in the computer vision community. The final object of SR is to restore a high-quality (HQ) image from its low-quality (LQ) observation, which suffers from various image quality degradations. The difficulty of SR mainly lies in two parts: (1) the unknown degradations (*e.g.*, blur, downsampling, noise, compression, and their combinations) of LQ. (2) the multiple solutions for a given LQ input image.

In recent years, numerous studies have been made to tackle this challenge, utilizing convolution neural networks (CNNs) (Dong et al., 2014; 2016; Ledig et al., 2017; Zhang et al., 2018c), vision transformers (ViTs) (Zhang et al., 2018b; Liang et al., 2021; Wang et al., 2022; Chen et al., 2022; 2023), and their combinations. Though achieving inspiring results, these methods mostly fail in real-world scenarios. This failure is attributed to assuming degradation as *an ideal bicubic downsampling kernel*, way too different from the unknown and complex degradation in real world. Therefore, it draws researchers' increasing attention to reconstruct perceptually realistic HQ images in real-world scenarios. Thereafter, this challenging and meaningful task is called real-world super-resolution (Real-SR) (Gu et al.,

2019; Zhang et al., 2021; Wang et al., 2021; Cai et al., 2019; Yu et al., 2024; Wu et al., 2024b; Sun et al., 2024).

Recently, diffusion models (DMs) demonstrate remarkable performance in image-generating tasks, particularly in perceptual quality. The excellence of DM comes from its vast prior knowledge in modeling real-world objects, especially generating clear textures and mitigating artifacts and distortions. The powerful realistic texture generation ability is inherently the same as the object of Real-SR problem, leading to plentiful breakthroughs (Yang et al., 2025; Yu et al., 2024; Rombach et al., 2022). However, the inference cost is still too high to run on edge devices. Therefore, it's essential to further compress DMs to accelerate the inference, reduce storage cost, and minimize degradation.

Popular model compression techniques include pruning, distillation, and quantization, among which, 1-bit quantization (*i.e.*, binarization) gains significant effectiveness. As an extreme quantization method, binarization compresses models' weights from 32-bit to only 1-bit, significantly reducing memory and computational cost. However, applying naive binarization will lead to catastrophic model collapse. Hence, additional structures are required to be designed.

A popular solution is adding full-precision information, *i.e.*, skip connection branch. However, due to UNet's frequent changes in resolution and dimension, skip connection faces the mismatch challenge. To address this problem, we propose two auxiliary branches, namely low rank matrix branch (LRMB) and sparse matrix branch (SMB). Inspired by LoRA, the proposed LRMB leverages low rank decomposition to achieve dimension shift. We select the top $r$ singular values in SVD and utilize its corresponding components to initialize the LRMB. As for SMB, we employ sparse matrix to absorb the top $k$ absolute values in a full-precision matrix. The weights of LRMB and SMB are subtracted from the binarized matrix branch (BMB) and it concentrates on restoring textures. Three branches form the BiMaCoSR and provide excellent performance shown in Fig. 1.

To sum up, the contributions of our work are as follows:

1. We design BiMaCoSR, a new binarized one-step diffusion model for image super-resolution. To the best of our knowledge, BiMaCoSR is the first binarized one-step diffusion model.

2. We propose LRMB, which leverages low rank decomposition and SVD initialization to carry low frequency information and decouple the effect of BMB.

3. We propose SMB, which utilizes sparse matrix compression and extreme value absorption to deliver high rank features and achieve further decoupling.

4. We conduct comprehensive comparison experiments to show the state-of-the-art performance of the proposed BiMaCoSR. Besides, extensive ablation studies are conducted to prove the robustness and efficacy.

## 2. Related Work

**Image Super-Resolution.** Deep learning based approaches have demonstrated striking power in the realm of SR (Dong et al., 2014; Luo et al., 2022; Wang et al., 2021; Lim et al., 2017; Chen et al., 2023). As a groundbreaking work, SRCNN (Dong et al., 2014) initiates the track of solving SR problem via deep learning based approach. Thereafter, substantial contributions have been made to explore the best SR network architecture. For example, RCAN (Zhang et al., 2018b) leverages the residual in residual structure and deepens the network to more than 400 layers. SwinIR (Liang et al., 2021) is based on vision transformer structure and utilizes spatial window self-attention to capture the overall structure information. CAT (Chen et al., 2022) combines the attention mechanism and the CNN structure to make the most of the local and the global information. However, most of these conventional image super-resolution methods can not handle the Real-SR task because of the complex degradation in the real world.

**Diffusion Model.** In recent years, the diffusion-based methods have gained remarkable performance in many computer vision tasks and SR is no exception. For instance, SR3 (Saharia et al., 2022) restores the LQ by transforming the standard normal distribution into the empirical data distribution by learning a series of iterative refinement steps. DiffBIR (Lin et al., 2024) capitalizes on two restoration stages to seek the tradeoff of fidelity and quality. SinSR (Wang et al., 2024) effectively reduces the inference step to only one step via distillation and regularization. Following SinSR, OSEDiff (Wu et al., 2024a) modifies the distillation paradigm, and novel losses are introduced to improve face restoration ability. Despite the greatly improved inference speed, the model size remains the sam,e and there is still room for further acceleration.

**Binarization.** As the most extreme form of quantization, binarization typically compresses the weight into only 1 bit. In binarization, all the weights are seen as $\pm 1$ and the multiplications between weights and activations are converted to bit operations on sign bit of activation, allowing maximum compression and acceleration. Binarization is mainly about classification tasks initially (Rastegari et al., 2016; Liu et al., 2020; Qin et al., 2020; 2022; Huang et al., 2024). Recently, researchers have begun to perform binarization on image restoration tasks. Binary Latent Diffusion (Wang et al., 2023b) trains an auto-encoder with a binary latent space and mainly focuses on the Bernoulli distribution instead of acceleration. BiDM (Zheng et al., 2024) leverages timestep-friendly binary structure and space-patched distillation to compress the diffusion model to 1 bit. BI-DiffSR (Chen et al., 2024) designs several binary friendly modules and redistributes the activation of different time step. However, the inference step remains the same. Therefore, it is necessary to further compress the model to one step.

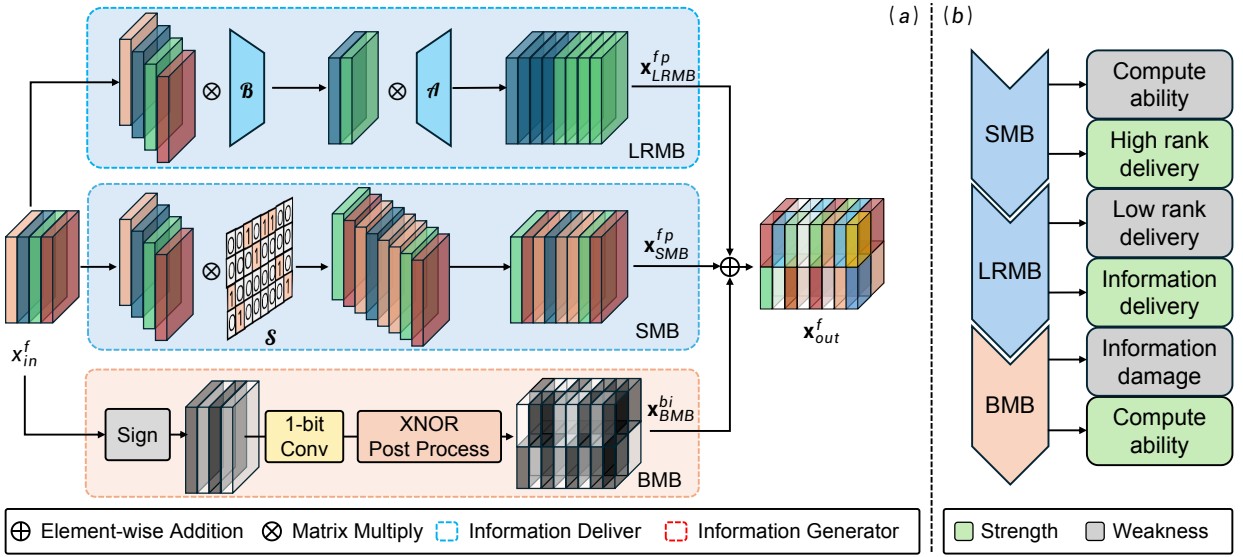

Figure 2: Overview of our proposed BiMaCoSR which employs three different compressed matrix branches. (a) The structure of a convolution layer in BiMaCoSR after binarization. Two auxiliary branches, *i.e.*, LRMB and SMB, support BiMaCoSR's excellent performance. The linear layer can be regarded as a $1 \times 1$ convolution layer and is processed with the same pipeline. (b) Illustration of the initialization and how the three branches solve the weakness of the other branches.

## 3. Methodology

In this section, we describe our proposed BiMaCoSR, shown in Fig. 2. First, we analyze three issues that have potential for further improvement when binarizing diffusion models. Thereafter, we describe our proposed branches, *i.e.*, low rank matrix branch (LRMB) and sparse matrix branch (SMB), which could serve as the auxiliary branch to binarized matrix branch (BMB). Finally, we illustrate the initialization methods, shown in Fig. 3, and why our designs work.

### 3.1. Analysis

Binarized blocks inevitably suffer from representation degradation due to the extreme 1-bit compression. Previous research made remarkable progress (Chen et al., 2024; Zheng et al., 2024) on binarization, yet there still remains room for improvement. We conclude three issues that deserve special attention in most diffusion models.

***Issue I: The specially severe degradation of linear layer.***
Experimentally, we observe that with same size of parameters, linear layer suffers from more severe degradation in comparison to convolution layer. This observation in SR is consistent with previous research (Le & Li, 2023). With the advancement of DiT and binarization, this issue is gradually becoming significant and an additional process to quantize or binarize the linear layer is in urgent need.

***Issue II: The frequent dimension changes.*** It's necessary to leverage skip connection as an auxiliary branch in binarized network. Skip connection could carry abundant feature information and bring negligible computation and storage overhead. However, in UNet, frequent changes in dimension

and resolution make two forms of skip connection (*i.e.*, addition and concatenation) not applicable. In addition, the distributions before and after binarized block are greatly different. Therefore, efficient module is critical to overcome the frequent changes and carry rich information.

***Issue III: The inadequate usage of pre-trained FP model.***
The pre-trained FP model is vital to quantization methods from any perspective. However, current QAT research does not make the most of the pre-trained model, simply loading the FP parameters and directly beginning the training. It is a popular way to leverage additional modules to enhance the binarized network. The impact of the modules could vary when different initialization methods are applied.

We conclude that the above issues are attributed to the insufficient ability of binarization. **To be specific, binarization is just one of the matrix compression methods, inherently unsuitable for multifaceted challenges.** Therefore, we propose following branches to compress matrix in different ways. With the combination of various compressed matrices, the issues above can be appropriately addressed.

### 3.2. Low Rank Matrix Branch

To address ***Issue I***, *i.e.*, the specially severe degradation of linear layer, we identify that the quantized linear layer is strongly limited by the bit-width. An FP linear layer usually serves two purposes, delivering the complete information of input and performing appropriate linear transformation. With limited bit-width, the quantized linear layer cannot play two roles at the same time. Therefore, we decouple the roles apart. Though binarized, the weight matrix is usually high rank and contributes more to high frequency, *i.e.*, the

detailed structure and texture in image. Hence, we design low rank matrix branch (LRMB) to serve as a complementary branch, transmitting low frequency information.

The idea of LRMB is inspired by low rank approximation in matrix theory, which is a common strategy for matrix compression. To transmit low frequency information, matrices with low rank are enough and efficient. Mathematically, given an $m \times n$ matrix $\mathbf{W}$, it can be approximated with two low rank matrices, *i.e.*, $\mathbf{W} \approx \hat{\mathbf{W}} := \mathbf{BA}$, where $\mathbf{B}$ and $\mathbf{A}$ are $m \times r$ and $r \times n$ matrices respectively, and $r \ll \min(m, n)$ is a hyper-parameters denoting the rank of $\hat{\mathbf{W}}$. Therefore, the formula of LRMB is:

$$\mathbf{x}_{\text{LRMB}} = \text{LRMB}(\mathbf{x}_{\text{in}}) := \mathbf{x}_{\text{in}}\mathbf{BA}, \qquad (1)$$

where $\mathbf{x}_{\text{in}} \in \mathbb{R}^{N \times m}$ and $\mathbf{x}_{\text{LRMB}} \in \mathbb{R}^{N \times n}$ are the input and output of $\text{LRMB}(\cdot)$ respectively, and $N$ is the number of tokens. Usually, $\mathbf{W}$ is a square matrix and $m = n$.

As for complexity, the storage overhead is:

$$O_s = (m \times r + r \times n)B = rB(m + n) \ll mnB', \quad (2)$$

where $B = 32$ or $16$ and $B' = 1$ are the number of bits required by one element in LRMB branch and binarized branch, respectively. Meanwhile, the computation overhead is:

$$O_c = N \times m \times r + N \times r \times n = Nr(m + n). \quad (3)$$

This means that even saved with 32 bits, the storage and computation overhead of LRMB is acceptable. In conclusion, with LRMB, *Issue I* and *Issue II* are partly solved.

### 3.3. Sparse Matrix Branch

LRMB is still not enough to replace the skip connection. This is because skip connection can be represented as a full rank identity matrix while LRMB is low rank. To compensate the missed ranks, we propose sparse matrix branch (SMB), which leverages sparse matrix compression and could efficiently deliver high rank information.

Sparse matrix is a matrix in which most of the elements are zero (Yan et al., 2017). A common criterion of a sparse matrix is that the number of non-zero elements is approximated to the number of rows or columns. One way to save a sparse matrix is via the coordinate format (COO), where the non-zero elements are represented by a list of triples and it component is $(row, col, value)$.

Specifically, to form a sparse matrix, we select $k$ critical values from the weight matrix and the selection method will be described in Sec. 3.4. In the forward process, the SMB process could be equivalently expressed as

$$\mathbf{x}_{\text{SMB}} = \text{SMB}(\mathbf{x}_{\text{in}}) := \mathbf{x}_{\text{in}}\mathbf{W}_{\text{sparse}}, \qquad (4)$$

where $\mathbf{W}_{\text{sparse}}$ is a matrix with $k$ non-zero elements. During training, the coordinates will be fixed while their values will be updated by gradient descent. With SMB, the high rank information could be delivered and the remaining parts of *Issue I* and *Issue II* could be solved.

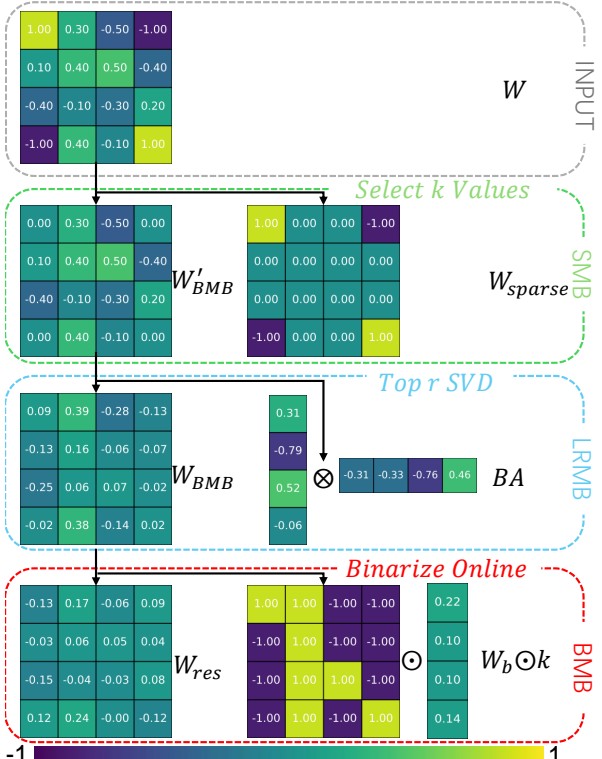

Figure 3: Initialization of different branch. $W_{res}$ represents the initial quantization error. In our method, $\|W_{res}\|_F^2 = 0.1855$, while $\|W_{res}\|_F^2 = 1.1275$ in direct binarization.

### 3.4. Pretrained-Friendly Initialization

We propose the following initialization method to guarantee better use of the pre-trained model, allowing better restoration performance and faster convergence.

The purpose of LRMB is to carry the low frequency information. Usually, compared with high frequency information, the magnitude of low frequency information is much greater. Therefore, we leverage SVD and select the top $r$ components to initialize LRMB. To be specific, we perform SVD on $\mathbf{W}$:

$$\mathbf{W} = \mathbf{U}\Sigma\mathbf{V}^T = \mathbf{U}\text{diag}\{\sigma_1, \sigma_2, \ldots, \sigma_n\}\mathbf{V}^T. \quad (5)$$

Then, we truncate the top $r$ singular values and absorb the remaining singular value into $\mathbf{U}$, forming matrices $\mathbf{A}$ and $\mathbf{B}$. The process can be written as:

$$\mathbf{W} \approx \underbrace{\mathbf{U}\text{diag}\{\sigma_1, \sigma_2, \ldots, \sigma_r, 0, \ldots, 0\}}_{B}\underbrace{\mathbf{V}^T}_{A} = \mathbf{BA}. \quad (6)$$

Thereafter, to guarantee that the binarized matrix only serves to add high frequency information, the low frequency counterpart will be subtracted. The formula is:

$$\mathbf{W}'_{\text{BMB}} := \mathbf{W} - \mathbf{BA}, \qquad (7)$$

where $\mathbf{W}'_{\text{BMB}}$ is the matrix to be binarized after the initialization of LRMB. Hence, the ability of weight matrix is decoupled into two components, the low frequency part ($\mathbf{BA}$) and the high frequency part ($\mathbf{W}'_{\text{BMB}}$).

Thereafter, the initialization of SMB also matters. As described in Sec. 3.3, we select $k$ values from the weight matrix to generate the sparse matrix. Due to the fixed coordinates of non-zero elements, random selection is apparently a good candidate. Besides, binarization often ignores the outlier values in weight matrix, which are often huge but rare (less than 0.1%). Though rare, these outliers play a crucial role in most models. Considering their rarity and significance, we select $k$ elements with the greatest absolute value in $\mathbf{W}'_{\text{BMB}}$. Subsequently, we subtract these values in $\mathbf{W}'_{\text{BMB}}$ to decouple the overlapping effect of BMB and SMB. The process can be formulated as:

$$\widetilde{w}_{ij} = \begin{cases} w_{ij}, & \text{if } |w'_{ij}| \geq t \\ 0, & \text{otherwise} \end{cases}, \mathbf{W}_{\text{BMB}} := \mathbf{W}'_{\text{BMB}} - \mathbf{W}_{\text{sparse}}, \tag{8}$$

where $\widetilde{w}_{ij}$ and $w_{ij}$ are the elements of $\mathbf{W}_{\text{sparse}}$ and $\mathbf{W}'_{\text{BMB}}$ respectively, $t$ is the $k$-th largest absolute value in $\mathbf{W}'_{\text{BMB}}$, $\mathbf{W}_{\text{BMB}}$ is the final matrix to be binarized.

### 3.5. Overall Structure

After adding LRMB and SMB and their initialization, the binarized matrix branch (BMB) is formed with $\mathbf{x}_{\text{BMB}}$. To be specific, the input feature and weight matrix will be binarized with $\text{Sign}(\cdot)$, which can be written as:

$$\begin{cases} \mathbf{x}_{\text{b}} = \text{Sign}(\mathbf{x}_{\text{in}}), \\ \mathbf{W}_{\text{b}} = \text{Sign}(\mathbf{W}_{\text{BMB}}), \end{cases} \text{Sign}(x) = \begin{cases} +1, & x \geq 0, \\ -1, & x < 0, \end{cases} \tag{9}$$

where $\text{Sign}(\cdot)$ is performed element-wise. Thereafter, the full-precision convolution and linear transform are replaced with efficient logical XNOR and bit-counting operations. The process can be formulated as:

$$\mathbf{x}'_{\text{BMB}} = \text{bit-count}(\text{XNOR}(\mathbf{x}_{\text{b}}, \mathbf{W}_{\text{b}})), \tag{10}$$

Finally, we follow the design of XNOR-Net (Rastegari et al., 2016) to compensate the precision loss, which is:

$$\mathbf{x}_{\text{BMB}} = \mathbf{x}'_{\text{BMB}} \odot (\mathcal{A} \otimes \mathbf{k}), \tag{11}$$

where $\mathcal{A}_{H \times W}$ is the channel-wise absolute average of input activation, $\mathbf{k}$ is a vector whose element is the absolute average of correspond channel, $\odot$ is Hadamard product, and $(\mathcal{A} \otimes \mathbf{k})_{c,i,j} := \mathcal{A}_{i,j}\mathbf{k}_c$ should be calculated first.

We binarized every convolution layer and linear layer except the first and last convolution layers due to extremely severe degradation of quantization on head and tail. Specifically, linear layer can be seen as the $1 \times 1$ convolution layer and therefore processed in the same way as convolution layer.

For one binarized layer, the output can be written as:

$$\mathbf{x}_{\text{out}} = \mathbf{x}_{\text{BMB}} + \mathbf{x}_{\text{LRMB}} + \mathbf{x}_{\text{SMB}}. \tag{12}$$

In conclusion, we propose LRMB, SMB, and their corresponding initialization methods. With these designs, our BiMaCoSR can significantly reduce the information loss caused by binarization and improve the restoration ability. Besides, the storage and computation overhead caused by LRMB and SMB can be safely ignored.

## 4. Experiments

### 4.1. Experimental Settings

**Data.** We take the models on the training set of ImageNet (Russakovsky et al., 2015) and the LR images are generated by the same pipeline of RealESRGAN (Wang et al., 2021). We evaluate the models with three benchmark datasets: RealSR (Cai et al., 2019), DRealSR (Wei et al., 2020), and DIV2K-Val (Agustsson & Timofte, 2017). RealSR and DRealSR are real world benchmarks while DIV2K-Val employs Bicubic interpolation to generate LR images. The upscale ratio of training set and test set is $\times 4$.

**Evaluation Metrics.** The evaluation metrics are implemented with IQA-Pytorch (Chen & Mo, 2022) and are twofold to assess the restoration and compression ability. **Firstly**, to thoroughly evaluate the restoration ability of our proposed BiMaCoSR, we employ the following quantitative metrics: full-reference metrics PSNR, SSIM (Wang et al., 2004), and LPIPS (Zhang et al., 2018a) and non-reference metrics DISTS (Ding et al., 2020), FID (Heusel et al., 2017), NIQE (Zhang et al., 2015), MANIQA-pipal (Yang et al., 2022), MUSIQ (Ke et al., 2021), and CLIPIQA+ (Wang et al., 2023a). PSNR and SSIM are distortion-based metrics and are calculated on the Y channel (*i.e.*, luminance) of the YCbCr space. The rest metrics are all perceptual metrics and it is widely known that perceptual metrics are more aligned with human when rating the image quality.

**Secondly**, to demonstrate binarization's extreme compression and acceleration ability, we use the total parameters and overall operations as key metrics. Following previous work (Xia et al., 2022; Qin et al., 2023), the total parameters (**Params**) of the model are calculated as Params=Params$^b$+Params$^f$, and the overall operations (**OPs**) as OPs=OPs$^b$+OPs$^f$, where Params$^b$=Params$^f$/32 and OPs$^b$=OPs$^f$/64; the superscripts $f$ and $b$ denote full-precision and binarized modules, respectively. The computational complexity is tested with the input size $3 \times 64 \times 64$.

**Implementation Details.** We take SinSR (Wang et al., 2024), a one-step diffusion model, as the backbone. We initialize with SinSR's weights and use ResShift (Yue et al., 2024) as the teacher model. All the rest convolution and linear layers, except the head and tail layers, are binarized into BMB to guarantee the best compression ratio and LRMB and SMB are attached with BMB. We set the rank of LRMB as $r = 8$ and the number of non-zero elements in SMB as $k = 2\max(C_1, C_2)$, where $C_1$ and $C_2$ are the numbers of input and output channels, respectively.

**Training Settings.** We utilize Adam optimizer (Kingma & Ba, 2015) with $\beta_1 = 0.9$ and $\beta_2 = 0.99$, and set the learning rate as $2 \times 10^{-5}$. The batch size is set to 8, with 100K iterations. The input LR images are center-cropped to size $64 \times 64$. All the training and testing experiments are conducted on one NVIDIA RTX A6000.

Table 1: Quantitative comparison with SOTA methods. We compare BiMaCoSR with full-precision models and current binarization methods. We employ 9 metrics commonly used in SR and test on three benchmarks. The best and second best value are marked with red and blue respectively. In conclusion, our proposed BiMaCoSR achieves SOTA performance.

| Datasets | Methods | Bits (W/A) | PSNR ↑ | SSIM ↑ | LPIPS ↓ | MUSIQ ↑ | MANIQA ↑ | DISTS ↓ | FID ↓ | NIQE ↓ | CLIP-IQA+ ↑ |
|---|---|---|---|---|---|---|---|---|---|---|---|
| RealSR | SinSR | 32/32 | 26.51 | 0.7380 | 0.3635 | 57.87 | 0.5139 | 0.2193 | 56.36 | 5.826 | 0.5736 |
| | ResShift | 32/32 | 25.45 | 0.7243 | 0.3731 | 56.23 | 0.5005 | 0.2344 | 58.14 | 7.353 | 0.5708 |
| | XNOR | 1/1 | 26.48 | 0.7434 | 0.3968 | 43.56 | 0.3732 | 0.2609 | 105.72 | 6.014 | 0.4380 |
| | ReActNet | 1/1 | 26.60 | 0.7530 | 0.3834 | 44.18 | 0.3829 | 0.2551 | 109.36 | 6.306 | 0.4361 |
| | BBCU | 1/1 | 26.43 | 0.7488 | 0.3902 | 43.70 | 0.3792 | 0.2575 | 108.32 | 6.058 | 0.4298 |
| | ReSTE | 1/1 | 26.26 | 0.7408 | 0.4184 | 41.04 | 0.3677 | 0.2719 | 113.86 | 6.174 | 0.4083 |
| | BiDM | 1/1 | 25.07 | 0.7036 | 0.5042 | 35.60 | 0.3517 | 0.3226 | 115.23 | 6.759 | 0.3935 |
| | BiMaCoSR | 1/1 | 26.84 | 0.7698 | 0.3375 | 49.01 | 0.4034 | 0.2183 | 86.09 | 5.856 | 0.4800 |
| DRealSR | SinSR | 32/32 | 27.89 | 0.7332 | 0.4499 | 30.81 | 0.4519 | 0.2209 | 16.56 | 5.789 | 0.6052 |
| | ResShift | 32/32 | 26.64 | 0.7298 | 0.4478 | 31.09 | 0.4345 | 0.2337 | 18.12 | 6.959 | 0.5795 |
| | XNOR | 1/1 | 29.03 | 0.8319 | 0.3712 | 26.19 | 0.3560 | 0.2447 | 29.88 | 6.229 | 0.4449 |
| | ReActNet | 1/1 | 29.34 | 0.8431 | 0.3571 | 26.83 | 0.3618 | 0.2411 | 30.18 | 6.561 | 0.4380 |
| | BBCU | 1/1 | 29.00 | 0.8385 | 0.3643 | 26.37 | 0.3594 | 0.2433 | 30.94 | 6.337 | 0.4383 |
| | ReSTE | 1/1 | 28.91 | 0.8353 | 0.3899 | 25.12 | 0.3509 | 0.2641 | 33.64 | 6.459 | 0.4131 |
| | BiDM | 1/1 | 27.40 | 0.7942 | 0.4849 | 23.38 | 0.3529 | 0.3118 | 37.83 | 6.753 | 0.4307 |
| | BiMaCoSR | 1/1 | 29.33 | 0.8393 | 0.3400 | 29.38 | 0.3802 | 0.2278 | 22.31 | 6.150 | 0.4867 |
| DIV2K-Val | SinSR | 32/32 | 27.75 | 0.7694 | 0.1903 | 64.62 | 0.5336 | 0.1029 | 6.27 | 4.308 | 0.6147 |
| | ResShift | 32/32 | 27.18 | 0.7667 | 0.1775 | 65.04 | 0.5548 | 0.1016 | 7.54 | 5.121 | 0.6280 |
| | XNOR | 1/1 | 26.44 | 0.7185 | 0.3727 | 49.10 | 0.3972 | 0.2204 | 55.77 | 5.320 | 0.4584 |
| | ReActNet | 1/1 | 26.49 | 0.7260 | 0.3602 | 50.29 | 0.4078 | 0.2111 | 52.32 | 5.366 | 0.4726 |
| | BBCU | 1/1 | 26.39 | 0.7221 | 0.3660 | 50.09 | 0.4035 | 0.2148 | 53.22 | 5.263 | 0.4653 |
| | ReSTE | 1/1 | 26.07 | 0.7125 | 0.3916 | 46.95 | 0.3907 | 0.2295 | 61.52 | 5.399 | 0.4328 |
| | BiDM | 1/1 | 24.29 | 0.6725 | 0.4370 | 40.15 | 0.3747 | 0.2916 | 62.28 | 6.090 | 0.4112 |
| | BiMaCoSR | 1/1 | 27.35 | 0.7547 | 0.2999 | 53.38 | 0.4337 | 0.1806 | 27.99 | 4.987 | 0.5176 |

## 4.2. Comparison with State-of-the-Art Methods

We compare our proposed BiMaCoSR with recent binarization methods, including XNOR (Rastegari et al., 2016), ReActNet (Liu et al., 2020), BBCU (Xia et al., 2022), ReSTE (Wu et al., 2023), and BiDM (Zheng et al., 2024). All binarization methods are implemented on SinSR (Wang et al., 2024) and trained with the same settings. As LRMB and SMB slightly increase the parameters, we additionally keep the first two and last two convolution layers as full-precision. We also compare BiMaCoSR with the full-precision model SinSR and ResShift (Yue et al., 2024). SinSR is the distilled version of ResShift. The comparison results are exhibited in quantitative and qualitative aspects.

**Restoration Results.** The quantitative comparison results are shown in Table 1. We can obtain the following observations. (1) The results in Table 1 demonstrate clear advantage over competing methods in both full-reference and non-reference metrics on three benchmark datasets. (2) In some situations, such as PSNR and LPIPS on RealSR, BiMaCoSR can even surpass SinSR and ResShift. We attribute the excellence to the LRMB and SMB, which transmit the full-precision information in a parameter-efficient way. (3) Most binarization methods outperform the baseline model and ReActNet presents competing performance, especially on DRealSR. (4) The full-precision models, *i.e.*, SinSR and ResShift, are excellent on perceptual metrics, such as DISTS and CLIP-IQA+. Whereas, the multi-step models'

performance on PSNR and SSIM is on the lower side. This phenomenon means that compression on DM leads to disappearance of texture and details. However, their performance on PSNR and SSIM is only slightly affected.

**Efficiency Comparison.** Efficiency comparison results are provided in Table 2. After the distillation from multi-step to one-step, the FLOPs are significantly reduced. Furthermore, compressing the model to 1 bit makes the model tiny and fast. In comparison with SinSR, the compression ratio is 27.45× and the speedup ratio is 23.81×. Compared with current SOTA binarization methods, we keep almost the same parameters but take much less FLOPs on one-step DM, due to the calculation efficient design. To conclude, our proposed BiMaCoSR gains remarkable compression ratio and speedup ratio and achieves outstanding performance.

**Visual Results.** We present visual comparison on challenging cases in Fig. 4. One typical challenging case is the tiny and dense structures, such as hairs, grass, tiles, and faces. It is struggling with previous binarization methods to restore image details, especially in challenging cases. On the contrary, our BiMaCoSR is able to recover results with sharper edges and richer textures. For example, in 0809, BiMaCoSR reconstructs the hairs on the nose while other methods just output rough color blocks. And in 0831, BiMaCoSR restores the woman's facial structures and expression while other methods smooth the organs to the same color. In 0834, BiMaCoSR could successfully recover the tiles' texture and

| | ResShift | SinSR | ReActNet | BBCU | ReSTE | BiDM | XNOR | Ours |
|---|---|---|---|---|---|---|---|---|
| Inference Step | 15 | 1 | 1 | 1 | 1 | 1 | 1 | 1 |
| FLOPs (G) | 753.45 | 50.23 | 5.83 | 5.83 | 5.83 | 11.60 | 5.83 | 1.83 |
| # Total Param (M) | 118.59 | 118.59 | 4.95 | 4.95 | 4.95 | 18.69 | 4.95 | 4.98 |
| PSNR/LPIPS | 25.45/0.3731 | 26.51/0.3635 | 26.60/0.3834 | 26.43/0.3902 | 26.26/0.4184 | 25.07/0.5042 | 26.48/0.3968 | 26.84/0.3375 |

Table 2: Efficiency comparison on RealSR. BiMaCoSR takes least FLOPs while gains the best performance.

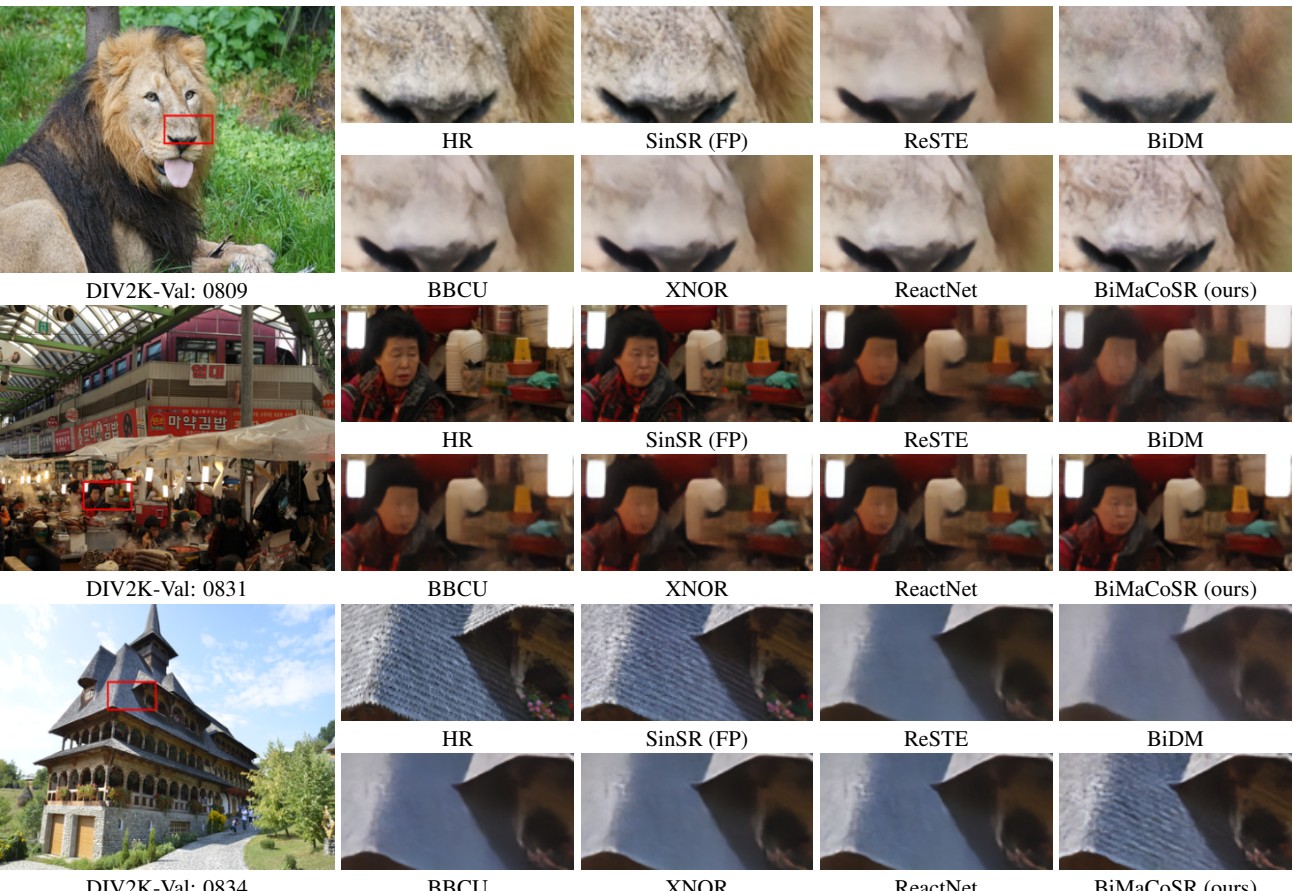

Figure 4: Visual comparison for image SR. We compare our proposed BiMaCoSR with current competitive binarization methods and the full-precision (FP) model. The visual results illustrate that BiMaCoSR gains rich details and reasonable textures.

most binarization methods fail. What's more, the difference between BiMaCoSR and the FP model (SinSR) exists but is minimal. To conclude, our proposed BiMaCoSR surpasses other methods according to visual comparison. Additionally, we provide more visual results and corresponding analysis in the supplementary material for further comparison.

### 4.3. Ablation Study

In this section, we adopt RealSR as the test set and other training settings are the same as Sec. 4.1. To test the effectiveness and robustness, we conduct five key ablation experiments, including the break down ablation, loss function, rank of low rank branch, initialization of LRMB, and initialization of SMB. What's more, detailed analysis is also provided in the following sections. These ablation studies demonstrate the robustness and efficiency of our LRMB, SMB, and their corresponding initialization methods.

**Break Down Ablation.** Table 3a shows results of the break down ablation. With only BMB, the model could be extremely compressed but the performance is somewhat on the low side. With LRMB, though the number of parameters increases to 1.83G, the performance also improves on all metrics and the FLOPs only increase to 4.98M. After adding SMB, the overhead of parameters and storage can be neglected. The performance on all perceptual metrics improves while both PSNR and SSIM drop slightly.

**Loss Function.** The loss functions of SinSR play a critical role in FP model. However, these loss functions are not suitable in the binarized situation. After removing unnecessary parts from the SinSR loss, we leave only the distillation loss and the results are shown in Table 3b. With only the distillation loss, BiMaCoSR gains improvement on both distortion metrics and perceptual metrics.

| Branch | PSNR ↑ | SSIM ↑ | LPIPS ↓ | MANIQA ↑ | FID ↓ | CLIP-IQA+ ↑ | FLOPs (G) | Param (M) |
|---|---|---|---|---|---|---|---|---|
| BMB | 26.41 | 0.7408 | 0.4141 | 0.3704 | 110.15 | 0.4325 | 0.78 | 3.69 |
| +LRMB | 26.95 | 0.7718 | 0.3400 | 0.3937 | 88.72 | 0.4663 | 1.83 | 4.98 |
| +LRMB+SMB | 26.84 | 0.7698 | 0.3375 | 0.4034 | 86.09 | 0.4800 | 1.83 | 4.98 |

(a) Break down ablation.

| Loss | PSNR ↑ | SSIM ↑ | LPIPS ↓ | CLIP-IQA+ ↑ |
|---|---|---|---|---|
| Distill loss | 26.83 | 0.7698 | 0.3375 | 0.4800 |
| SinSR loss | 26.37 | 0.7466 | 0.4029 | 0.4273 |

(b) Ablation study on losses.

| Initialization | PSNR ↑ | SSIM ↑ | LPIPS ↓ | CLIP-IQA+ ↑ |
|---|---|---|---|---|
| Zero + Random | 26.88 | 0.7660 | 0.3497 | 0.4674 |
| SVD | 26.84 | 0.7698 | 0.3375 | 0.4800 |

(c) Ablation study on LRMB initialization.

| Rank | PSNR ↑ | SSIM ↑ | LPIPS ↓ | CLIP-IQA+ ↑ | FLOPs (G) | Param (M) |
|---|---|---|---|---|---|---|
| 4 | 26.29 | 0.7383 | 0.4197 | 0.4411 | 1.31 | 4.37 |
| 8 | 26.84 | 0.7698 | 0.3375 | 0.4800 | 1.83 | 4.98 |
| 12 | 26.97 | 0.7695 | 0.3400 | 0.4766 | 2.32 | 5.60 |
| 16 | 26.72 | 0.7625 | 0.3442 | 0.4971 | 2.83 | 6.21 |

(d) Ablation study on the rank of LRMB.

| Initialization | PSNR ↑ | SSIM ↑ | LPIPS ↓ | CLIP-IQA+ ↑ |
|---|---|---|---|---|
| Zero initial | 26.72 | 0.7628 | 0.3561 | 0.4639 |
| Uni-shortcut | 25.69 | 0.7113 | 0.5105 | 0.4008 |
| Sparse skip | 26.84 | 0.7698 | 0.3375 | 0.4800 |

(e) Ablation study on SMB initialization.

Table 3: Ablation studies on branches, loss, rank of LRMB and initialization of LRMB and SMB. The experiments is tested on RealSR. The comprehensive results demonstrate the robustness and efficient performance of our proposed BiMaCoSR.

**Rank of LRMB.** The rank of LRMB significantly influences the complexity, while the performance is not consistent with the increase in complexity. As shown in Table 3d, the model with rank of 16 performs best only on CLIP-IQA+, while the model with rank of 8 performs best and takes acceptable parameters. This is because the LRMB with high rank may influence high frequency generated by BMB, making the training unstable and leading to worse results. Therefore, we ultimately set the rank of LRMB to 8.

**Initialization of LRMB.** Conventional LoRA's initialization method is set the first matrix with random values and the second matrix with zero values. We compare this initialization method with our proposed SVD initialization and the result is shown in Table 3c. To conclude, our method outperforms conventional methods on SSIM, LPIPS, and CLIP-IQA+. We attribute it to the decoupling of transmitting low frequency information and generating high frequency. To conclude, our proposed SVD allows better performance.

**Initialization of SMB.** We search three popular ways to initialize SMB, including (1) random position and zero initialization, (2) Uni-shortcut (Xu et al., 2022), and (3) our proposed sparse skip. The experiment results in Table 3e show that our sparse skip initialization consistently enjoys the best performance on all four metrics. This improvement compared with other two initialization methods strongly supports the effectiveness of decoupling and absorption.

### 4.4. Visualization Analysis.

In Sec. 3, we assume that LRMB and SMB are in charge of transmitting low frequency information while BMB generates high frequency after decoupling. To validate this assumption, we visualize the proportion of high frequency generated by three branches in first 50 Conv layers and the result is shown in Fig. 5. On average, high frequency infor-

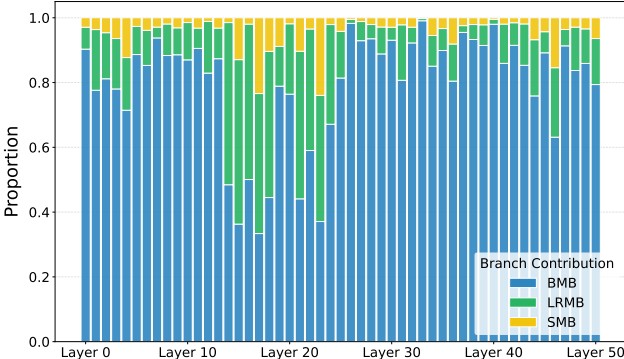

Figure 5: The proportion of high frequency information generated by three branches. The high frequency information mainly comes from BMB, which obeys our assumption.

mation is mainly generated by BMB, accounting for around 70%. With more parameters, LRMB also provides high frequency information in some layers but overall proportion is relative low. SMB only absorbs the extreme values and indeed carries little high frequency information. Fig. 5 shows that division of work is clear and our designs in LRMB, SMB and BMB obey our assumption.

## 5. Conclusion

In this paper, we propose the BiMaCoSR, a binarized SR diffusion model with only one inference step. Detailedly, we first propose LRMB and SVD initialization to decouple the effect of binarized branch and deliver low frequency information. Furthermore, we propose SMB and sparse initialization to absorb the extreme values and provide high rank representations. Comprehensive comparison experiments demonstrate the SOTA restoration ability of the proposed BiMaCoSR. Extensive ablation studies exhibit the efficiency and robustness of both LRMB and SMB. In the future, we will focus on the combination of pruning and binarization on one-step diffusion models for further compression.

## Acknowledgements

This work was supported by Shanghai Municipal Science and Technology Major Project (2021SHZDZX0102), the Fundamental Research Funds for the Central Universities, and the National Natural Science Foundation of China (NSFC No. W2412089).

## Impact Statement

This paper presents work whose goal is to advance the field of Machine Learning. There are many potential societal consequences of our work, none of which we feel must be specifically highlighted here.

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
