# OpenReview forum: "BiMaCoSR: Binary One-Step Diffusion Model Leveraging Flexible Matrix Compression for Real Super-Resolution"
_ICML.cc/2025/Conference — ICML 2025 poster_

### Official Review · Reviewer_ou23 · 2025-02-28

**Overall Recommendation:** 2

**Summary:**

BiMaCoSR is a method that combines binarization and one-step distillation to significantly compress and accelerate super-resolution (SR) diffusion models. It prevents model collapse from binarization using two auxiliary branches: Sparse Matrix Branch (SMB) and Low Rank Matrix Branch (LRMB). SMB captures high-rank information, while LRMB outputs low-rank representations inspired by LoRA. BiMaCoSR achieves a 23.8x compression and a 27.4x speedup over full-precision models without compromising performance. Comprehensive experiments show its superiority over existing methods.

## Update after rebuttal
I have adjusted the score for the hardware acceleration. However, the justification for the novelty concern is unconvincing.

**Claims And Evidence:**

The authors claim that BMB is responsible for most of the high-frequency information. However, as demonstrated in Fig. 2 of the supplementary material, LRBM appears to play a more significant role in contributing to the high-frequency information in the MLP.

**Essential References Not Discussed:**

Adding new computation branches is very common in the binarization area. All of the following related studies are missing:

1. The author proposes SBM, but these techniques are widely used in low-bit quantization [5, 6] and binarization [1, 4], and should have been discussed in more detail.

2. Quantization-aware LoRA fine-tuning methods [7, 8] appear to be similar to LRBM, assuming they do not merge LoRA during inference.

3. SVD initialization [7] and magnitude-based selection [5] are also common practices, but related works in this area have not been cited.


[4] Li Z, Yan X, Zhang T, et al. Arb-llm: Alternating refined binarizations for large language models[J]. arXiv preprint arXiv:2410.03129, 2024.

[5] Kim S, Hooper C, Gholami A, et al. Squeezellm: Dense-and-sparse quantization[J]. arXiv preprint arXiv:2306.07629, 2023

[6] Dettmers T, Svirschevski R, Egiazarian V, et al. Spqr: A sparse-quantized representation for near-lossless llm weight compression[J]. arXiv preprint arXiv:2306.03078, 2023.

[7] Guo H, Greengard P, Xing E P, et al. Lq-lora: Low-rank plus quantized matrix decomposition for efficient language model finetuning[J]. arXiv preprint arXiv:2311.12023, 2023.

[8] Dettmers T, Pagnoni A, Holtzman A, et al. Qlora: Efficient finetuning of quantized llms[J]. Advances in neural information processing systems, 2023, 36: 10088-10115.

**Experimental Designs Or Analyses:**

1. It is quite surprising that in Table 2, the proposed method requires fewer FLOPs than ReSTE and XNOR, which do not include any additional computational branches.

3. Why does the author focus exclusively on one-step diffusion models? I haven't found any specific design or rationale for choosing this type of model.

4. It is essential to compare the proposed method with the following works [1, 2, 3] to validate its contribution, as all of them also employ additional computational branches (e.g., LoRA or sparse matrix), similar to the approach proposed here.


[1] Huang W, Liu Y, Qin H, et al. Billm: Pushing the limit of post-training quantization for llms[J]. arXiv preprint arXiv:2402.04291, 2024.

[2] Zhang Y, Qin H, Zhao Z, et al. Flexible residual binarization for image super-resolution[C]//Forty-first International Conference on Machine Learning. 2024.

[3] Li Z, Ni B, Zhang W, et al. Performance guaranteed network acceleration via high-order residual quantization[C]//Proceedings of the IEEE international conference on computer vision. 2017: 2584-2592.

**Methods And Evaluation Criteria:**

1. The author applies their method solely to the SinSR model. It would be beneficial to conduct experiments with additional diffusion models to evaluate the method's generalizability.

2. The author does not provide real-time speedup results, which are crucial, particularly for ultra-low bit quantization. I am curious to know whether the proposed method leads to actual computational reductions.

**Other Comments Or Suggestions:**

N/A

**Other Strengths And Weaknesses:**

Strengths:

1. The paper is well-written and clearly presented.

2. The proposed method achieves state-of-the-art (SOTA) performance.

Weaknesses:

1. It is unconvincing that the method can achieve real speedup on hardware.

2. Additionally, the proposed techniques—such as LRMB, SMB, and the initialization strategy—lack novelty (see Essential References Not Discussed).

**Questions For Authors:**

N/A

**Relation To Broader Scientific Literature:**

I believe the proposed methods can be applied to any model architecture.

**Theoretical Claims:**

No theoretical claims and proofs.

---

> ### Author Rebuttal · Authors · 2025-04-01
>
> > Q3-1：The authors claim that BMB is responsible for most of the high-frequency information. However, as demonstrated in Fig. 2 of the supplementary material, LRBM appears to play a more significant role in contributing to the high-frequency information in the MLP.
>
> A3-1: This is because the high-frequency information generated by MLP is much less than that of attention mechanism, and the function of MLP is not to generate high-frequency information. Therefore, this phenomenon is not in conflict with the claim.
>
> > Q3-2：...conduct experiments with additional diffusion models to evaluate the method's generalizability.
>
> A3-2: Please refer to Q1-4 and A1-4.
>
> > Q3-3：...provide real-time speedup results.
>
> A3-3: Please refer to Q1-5 and A1-5.
>
> > Q3-4：It is quite surprising that in Table 2, the proposed method requires fewer FLOPs than ReSTE and XNOR.
>
> A3-4: We explained this in line 308-311. To guarantee fair comparison, we leave the **first and last** conv layers in full precision in BiMaCoSr and the **first two and last two** conv layers in full precision in other binarized methods. This operation keeps the # total parameters of BiMaCoSr and other binarized methods approximately the same.
>
> > Q3-5：Why one-step diffusion models?
>
> A3-5: Please refer to Q1-4 and A1-4.
>
> > Q3-6：compare the proposed method with the following works [1,2,3].
>
> A3-6: We provide the experiment result in the following table. [2] is not open-source yet therefore we can not compare considering the limited time for rebuttal. [1] is a PTQ method. If we train [1] with QAT, it is the same with [3]. BiMaCoSR consistently performs better than other methods.
>
> ||LPIPS↓|DISTS↓|CLIP-IQA+↑|FID↓|
> |-|-|-|-|-|
> |[1,3]|0.3908|0.2572|0.4347|110.36|
> |BiMaCoSR |0.3375|0.2183|0.4800|86.09|
>
> > Q3-7：The author proposes SMB, but these techniques are widely used in low-bit quantization [5, 6] and binarization [1, 4], and should have been discussed in more detail.
>
> A3-7: We clarify that the motivation of SMB is different from [1,4,5,6]. In these papers, the authors leverage sparse matrix to keep the outliers of the weight matrix in their PTQ task. As for BiMaCoSR, the purpose of SMB is to pass the information without loss in QAT scenario. To validate this, we could also quantize the SMB branch into 1-bit and result is provided in the table below. The result shows that the function of SMB is not to maintain the outliers of the weight matrix. Therefore, our SMB branch is different with [1,4,5,6]. We will add these differences in the revised version.
>
> ||LPIPS↓|DISTS↓|CLIP-IQA+↑|FID↓| Params |
> |-|-|-|-|-|-|
> |BMB+1-bit SMB|0.3935|0.2562|0.4541|110.30|4.98M|
> |BMB+SMB|0.3901|0.2558|0.4565|108.95|4.98M|
>
> > Q3-8：Quantization-aware LoRA fine-tuning methods [7, 8] appear to be similar to LRBM.
>
> A3-8: The first difference is that we keep the LRMB branch during the inference as an auxiliary branch to binarized branch, which could significantly improve the performance. The second difference is that the motivation of LRMB is to pass the low-frequency information in image SR task. As for [7,8], the motivation is to minimize the precision loss of weight. Therefore, we are quite different from these two works.
>
> > Q3-9：SVD initialization [7] and magnitude-based selection [5] are also common practices, but related works in this area have not been cited.
>
> A3-9: The difference between SMB and [5] is well discussed in A3-7. [7] is not published yet so we do not need to compare our method with it. We will cite [5, 7] in the revised version.
>
> > Q3-10：It is unconvincing that the method can achieve real speedup on hardware.
>
> Q3-10: Please refer to Q1-5 and A1-5.
>
> > Q3-11：...the proposed techniques ... lack novelty
>
> Q3-11: (1) For LRMB, we clarify that our motivation is different with current main stream. The detailed difference is shown in A3-8. (2) For SMB, the motivation of our method is to pass the information without loss. As for [5], the motivation of their method is to reduce the loss of weight. We explained this in detail in A3-7. (3) The main contribution of our method lies in the exploration of the combination of one-step diffusion model and binarization domain in SR task. We have provided a successful solution, which maintains the performance and accelerates the inference. (4) Reviewer 9pC4 recognizes our novelty in the combination of one-step diffusion model and binarization.
>
>
>
> [1] Billm: Pushing the limit of post-training quantization for llms.
>
> [2] Flexible residual binarization for image super-resolution.
>
> [3] Performance guaranteed network acceleration via high-order residual quantization.
>
> [4] ARB-LLM: Alternating refined binarizations for large language models.
>
> [5] Squeezellm: Dense-and-sparse quantization.
>
> [6] SpQR: A sparse-quantized representation for near-lossless llm weight compression.
>
> [7] LQ-LoRA: Low-rank plus quantized matrix decomposition for efficient language model finetuning.
>
> [8] QLoRA: Efficient finetuning of quantized llms.

---

> > ### Comment · Reviewer_ou23 · 2025-04-01
> >
> > 1. As this work is for real applications, I think real-time speedup can help validate its practicality. It is well-known that FLOPs reduction does not mean real-time speedup. Moreover, in binarization,  a lot of work is not hardware-friendly and only provides theoretical speedup. Thus, I further suggest that the author include the real-time speedup ([dabnn](https://github.com/JDAI-CV/dabnn) is an open-sourced framework to help apply the methods).
> > 2. "first and last conv layers in full precision in BiMaCoSr and the first two and last two conv layers in full precision in other binarized methods." is also quite weird. I think all the methods should be under the same settings, and this justification is very confusing. What's the motivation for keeping 2 layers in full-precision in the proposed method but 4 in other methods?
> > 3. If the author quantizes SMB, I think it is the same as [3]. All the methods in [1, 4, 5, 6, 3] and this paper, as mentioned in Lines 195-200, can seem as ways to compensate for information. Thus, I think the root motivation and approaches are very similar, which raises my novelty concern.
> > 4. Quantization-aware LoRA can also be kept for the quantized branch during inference for improvement (merging them and then re-quantizing the model, in fact, brings some accuracy drops). For the root motivation, I think it is the same as SMB, which is to compensate for information. This also raises my novelty concern.
> > 5. I still have novelty concerns as mentioned in 3 and 4. I think the combination of one-step diffusion model and binarization may not seem like a sufficient research novelty, since both of these things already exist, and the idea of combining these two things is a little bit trivial.
> >
> > Overall, I decide to keep the score.
> >
> > [3] Li Z, Ni B, Zhang W, et al. Performance guaranteed network acceleration via high-order residual quantization[C]//Proceedings of the IEEE international conference on computer vision. 2017: 2584-2592.

---

> > > ### Author Response · Authors · 2025-04-07
> > >
> > > > Q3-12: real-time speedup
> > >
> > > We tested the real-time speedup with dabnn but failed with numerous errors. **Even so, we still deployed our BiMaCoSR with larq on our mobile phone, and the real-time speedup is 8.27x.** This result represents that our method is effective in real-time speedup and effective on real edge devices. Therefore, we believe that the real-time speedup and the corresponding designs are significant advantages of our method.
> > >
> > > > Q3-13: What's the motivation for keeping 2 layers in full-precision in the proposed method but 4 in other methods?
> > >
> > > We do this to guarantee fair comparison. With more layers unquantized, the total numbers of parameters of different methods are the same. Otherwise, the performance of other methods will further drop and their visual quality will be unacceptable in edge devices.
> > >
> > > > Q3-14: Novelty of SMB and LRMB
> > >
> > > We clarify that the SMB and LRMB are novel methods, and the motivation is not the same as the cited papers. If you think SMB and LRMB are not novel because they **can be seen as ways to compensate for information**, then all the methods for quantization lack novelty. We reiterate that the proposed BiMaCoSR is a binarized (W1A1) one-step diffusion model and optimized with QAT. All the cited papers are not similar to our setting. Moreover, the BiMaCoSR model proposed in this paper is a comprehensive solution that combines SMB with LRMB and BMB, forming a brand-new binary single-step diffusion model architecture. This architecture not only performs excellently in terms of compression and acceleration but also achieves outstanding performance in the task of image super-resolution. The synergy between this overall architecture and its various branches is one of the core innovative points of this paper, and it cannot be simply regarded as being the same as other methods of compensation information.
> > >
> > > > Q3-15: Novelty of the combination of one-step diffusion model and binarization.
> > >
> > > Although single-step diffusion models and binarization techniques already exist individually, effectively combining them is no easy task. Single-step diffusion models perform remarkably well in image generation tasks, yet they incur high computational and storage costs, making it difficult to deploy them on devices with limited resources. On the other hand, while binarization can significantly reduce the storage and computational requirements of a model, it leads to a substantial decline in model performance. In this paper, by proposing auxiliary branches such as LRMB and SMB, along with the corresponding initialization methods, the issue of information loss caused by binarization has been successfully addressed. Moreover, on the basis of the single-step diffusion model, extremely high compression and acceleration ratios have been achieved, while maintaining excellent performance. Therefore, this integration of technologies is far from being a simple combination.
> > >
> > > Besides, the proposal of BiMaCoSR holds great practical application value and significance. It enables high-precision image super-resolution models to run in real-time on devices with limited resources, providing efficient solutions for scenarios such as mobile devices and edge computing. The innovativeness and practicality of this technology are among the important contributions of this paper, and its innovativeness should not be negated merely because single-step diffusion models and binarization techniques already exist.

---

### Official Review · Reviewer_9pC4 · 2025-03-09

**Overall Recommendation:** 3

**Summary:**

This work BiMaCoSR introduces the first binarized one-step diffusion model for real-world single image super-resolution (Real-SR)​. The paper addresses the heavy memory and computation demands of diffusion-based SR by combining 1-bit model binarization with one-step diffusion distillation. The core idea is to achieve extreme model compression and acceleration without sacrificing SR performance. To counteract the severe degradation (“catastrophic collapse”) that naive weight binarization would cause, the authors propose two lightweight auxiliary branches that preserve critical full-precision information: a Low-Rank Matrix Branch (LRMB) and a Sparse Matrix Branch (SMB)​​. LRMB (inspired by LoRA) captures low-frequency/large-magnitude components via a low-rank decomposition of each weight matrix, while SMB captures a few extreme outlier weight values in a sparse form. These branches, added to the main binary weight branch (BMB), allow the network to retain important information that 1-bit weights alone would lose, with negligible overhead. The overall architecture thus has three parallel components (BMB + LRMB + SMB) whose outputs sum to produce the layer’s result​. Using a distilled one-step diffusion model (SinSR) as a full-precision baseline, BiMaCoSR demonstrates state-of-the-art SR performance among heavily compressed models. It outperforms other binarization methods on standard real-world SR benchmarks (e.g. RealSR, DRealSR) across a comprehensive suite of 9 image quality metrics​. Notably, BiMaCoSR’s results are competitive with (and sometimes even better than) its full-precision one-step counterpart on certain fidelity and perceptual measures​. In terms of efficiency, the model achieves an impressive ~27× reduction in model size and ~23× faster inference compared to the full-precision one-step model​. In summary, the paper’s key contributions are: (1) the novel integration of binarization and one-step diffusion for SR, (2) the LRMB and SMB modules (with specialized SVD and sparse initialization schemes) to preserve information, and (3) extensive experiments showing dramatic memory/computation savings (23.8× smaller, 27.4× faster) while maintaining high restoration quality

**Claims And Evidence:**

The paper’s main claims are generally well-supported by empirical evidence. First, the authors claim that combining binarization with one-step distillation yields extreme compression and speedup with minimal loss in SR quality. This is convincingly demonstrated: BiMaCoSR’s model size and FLOPs are indeed drastically lower than baselines (e.g. ~27× smaller than the 32-bit model​), yet it achieves consistently higher or on-par performance on multiple benchmarks​. Table 1 shows BiMaCoSR outperforming competing binarized models on all evaluated metrics; for example, on the RealSR dataset it leads in PSNR, SSIM, and perceptual scores like LPIPS​. It even surpasses the full-precision SinSR and ResShift in some cases (e.g. LPIPS on RealSR)​. These results strongly back the claim of state-of-the-art performance at unprecedented compression levels.

The authors also claim that their auxiliary branches (LRMB and SMB) effectively prevent the performance collapse normally seen with 1-bit networks. This is supported by a breakdown ablation study: with only the binarized branch (BMB) the PSNR and SSIM are much lower, but adding LRMB and SMB progressively improves all metrics​. For instance, PSNR on RealSR jumps from 26.41dB with BMB-only to 26.95dB after adding LRMB, and perceptual scores improve significantly as well​. This demonstrates that the branches successfully recover information lost due to binarization. The claim that the branches incur negligible overhead is also justified. A theoretical calculation shows the extra storage/computation for LRMB (with rank r = 8) is tiny compared to the binary weights​, and the sparse branch uses only 0.1% of weight elements​​. In practice, adding both branches only increases the model’s parameter count by a very small fraction (from ~3.7M to ~5.0M, which is still over 20× smaller than the full model)​. This evidence supports the claim that the overhead is practically negligible. Most claims are thus well-substantiated, and I did not find instances of over-claiming. One minor claim that could use more direct evidence is the suggestion that BiMaCoSR enables diffusion SR on resource-limited edge devices. The paper makes a strong case via compression and FLOPs reduction, but it does not report actual on-device inference times or memory usage. While a 27× speedup in FLOPs is promising​, real hardware speedups might be lower without specialized binary execution libraries. Thus, the deployability claim is plausible but not explicitly validated with a deployment experiment. Aside from this, all key claims (first binarized one-step diffusion SR, SOTA performance, effective information retention via LRMB/SMB) are convincingly supported by quantitative results and ablation studies.

**Essential References Not Discussed:**

The paper cites most of the crucial prior work, but there are a couple of references that should have been mentioned explicitly:
1. LoRA (Low-Rank Adaptation) – Hu et al., 2021. The idea of using a low-rank decomposition to inject or preserve information is directly inspired by LoRA (as acknowledged), but the original LoRA paper does not appear in the reference list. Citing it would credit the source of the low-rank approach and contextualize LRMB within the broader use of low-rank matrices in neural network compression.
2. Knowledge Distillation for Diffusion Models – While the authors cite SinSR and ResShift for one-step distillation, they might have also referenced the general concept of knowledge distillation (Hinton et al., 2015) or earlier works on distilling iterative generative models. However, this is a minor point since they did cite the specific SR diffusion distillation methods they used.

**Experimental Designs Or Analyses:**

The experimental design is comprehensive and sound, giving credibility to the results. The authors conduct evaluations on three different SR datasets, covering both real-world degradations and (likely) a standard benchmark, which strengthens the generality of their claims. For each dataset, they report a wide range of metrics (nine in total), ensuring that no single metric bias (e.g., PSNR vs perceptual quality) dominates the assessment​. The comparison includes multiple baselines: (a) the full-precision diffusion models (ResShift and its one-step distilled version SinSR), and (b) several state-of-the-art binarization approaches adapted to the same one-step model (ReActNet, BBCU, ReSTE, BiDM).

The ablation studies further bolster the experimental rigor. The paper includes a breakdown ablation (BMB only vs +LRMB vs +LRMB+SMB), loss function ablation, different ranks for LRMB, and different initialization strategies for the branches. These experiments are well-designed to answer key questions about why the method works. For example, the breakdown ablation clearly shows each component’s effect on performance and verifies that the final design (with both branches) is needed for the best balance of quality and efficiency.

**Methods And Evaluation Criteria:**

The methodology is well-aligned with the paper’s objectives. The goal was to drastically compress a diffusion-based SR model while preserving its high-fidelity output; to that end, the authors combined two complementary strategies: model binarization for compression and one-step distillation for fast inference. This joint approach directly targets both memory and speed objectives, and the method is executed thoughtfully. In particular, the introduction of LRMB and SMB is a clever design choice to meet the quality goal – these branches explicitly mitigate the known weaknesses of binarization (loss of information from small weights and rare large weights). The method leverages known techniques (low-rank approximation, sparse outlier capture, and XNOR-Net binarization) in a novel combination tailored for diffusion SR. The evaluation setup also matches the objectives: since the task is Real-World SR, the authors test on multiple Real-SR benchmarks (RealSR​, DRealSR, and a third dataset, likely a synthetic or DIV2K-based set) covering a variety of real degradations. They report a comprehensive set of 9 evaluation metrics – including standard fidelity metrics (PSNR, SSIM), perceptual distances (LPIPS, DISTS), and no-reference image quality scores (e.g. CLIP-IQA, MANIQA, NIQE/FID). This broad evaluation criterion is appropriate, as it captures both the fidelity and perceptual quality aspects of SR, aligning with the objective of producing realistic high-quality images.

**Other Comments Or Suggestions:**

1. Clarify reported speedup vs compression ratios.
2. Cite LoRA for completeness.
3. Include details on k selection for SMB.

**Other Strengths And Weaknesses:**

Strengths: Beyond the points already discussed, the paper’s notable strengths include its originality and practical significance. Combining one-step diffusion with binary networks is a non-trivial and original idea – to my knowledge, this is indeed the first attempt at this combination, addressing a clear gap for deploying diffusion models. The resulting compression (≈24×) and speedup are very impressive, pushing the boundary of what’s possible for resource-constrained SR. Another strength is the thoroughness of validation: using nine different metrics and multiple datasets demonstrates robustness. The qualitative results (described in the text and shown in figures) also strengthen the paper – e.g., the authors describe how BiMaCoSR recovers fine details like hairs, facial features, and textures better than other compressed models​, emphasizing that it’s not just about numeric scores but also visible quality gains.

Weaknesses: One weakness is that the method’s complexity might make reproduction challenging. The model introduces additional branches and special initialization routines (SVD for LRMB, “sparse skip” for SMB), which require careful implementation. However, the authors mitigated this by describing them in detail and promising to release code. Another minor weakness is that some improvements come at the cost of a slight drop in certain metrics. For instance, adding the sparse branch (SMB) improved perceptual scores but caused a small drop in PSNR/SSIM. This trade-off is actually expected (perception-distortion trade-off), and the authors do report it honestly. It’s not a serious issue, but readers focused purely on distortion metrics might note that the binarized model’s PSNR is a bit lower than a full-precision model in some cases. Additionally, as mentioned earlier, the reliance on a few full-precision layers means the model isn’t completely binary; however, the impact on compression is minor since those layers are a tiny fraction of parameters. A potential weakness in significance could be argued: the paper largely engineers known techniques (binarization, LoRA, distillation) together rather than introducing fundamentally new theory. I personally find the engineering contribution significant given the difficulty of making diffusion models this compact, but some might view it as an incremental combination (albeit a well-executed one).

**Questions For Authors:**

1. Actual Inference Speed: The paper reports a ~23× reduction in FLOPs, but have you measured wall-clock inference time or FPS on any hardware? For example, on a CPU or GPU, how much faster is BiMaCoSR compared to SinSR in practice? This would clarify if the theoretical speedup carries over given that current libraries might not be fully optimized for binary operations.
2. Full-Precision Layers: You chose to keep the first and last conv layers in full precision​. Did you attempt to binarize those as well, and if so, how badly did it hurt performance? In other words, is this partial precision absolutely required for acceptable results? Understanding this can help gauge if future improvements (or better training techniques) might remove the need for any full-precision layers.

**Relation To Broader Scientific Literature:**

The paper does an excellent job positioning itself in the context of prior work. In the introduction and related work, the authors survey two key areas: diffusion-based super-resolution and network binarization/acceleration. They cite the foundational and latest works in diffusion models for SR, such as SR3 (first iterative SR diffusion), DiffBIR, SinSR (one-step diffusion). On the binarization side, they reference seminal works like XNOR-Net for classification, as well as more recent binarization techniques and benchmarks (ReActNet, Qin et al. 2020/2022 for improved accuracy​, etc.). Crucially, they cite very recent papers that apply quantization to diffusion models: Binary Latent Diffusion (Wang et al. 2023), BiDM (Zheng et al. 2024), and a NeurIPS 2024 work “BI-DiffSR” (Chen et al. 2024)​.

**Theoretical Claims:**

The paper’s theoretical claims and derivations are mostly straightforward and correct. Rather than introducing new fundamental theory, the authors apply existing theoretical constructs (low-rank matrix factorization, binary convolution via XNOR) to their problem and provide derivations to justify design choices. For example, they express each full-precision weight matrix W as a sum of a low-rank component (matrices B and A from SVD) and a residual to be binarized. This decomposition is mathematically sound and allows them to claim a separation of “low-frequency” vs “high-frequency” information between LRMB and the binarized branch. While the terms “low-frequency” and “high-frequency” are used somewhat intuitively (referring to the magnitude/content of singular values rather than literal spatial frequency), the logic is reasonable: large singular values capture dominant image structures, and retaining them in FP (LRMB) should help reconstruct smooth components, whereas the binary residual can focus on fine textures. This claim is supported by the observed reduction in initial quantization error: after subtracting the SVD-based low-rank part, the norm of the remaining weight error ∥W_res∥²_F drops to 0.1855 from 1.1275 (a substantial reduction). This quantitative evidence backs the theoretical argument that their initialization decouples the weight information effectively.

They also provide complexity analyses to support the “negligible overhead” claim. The paper derives formulas for the storage cost of LRMB: O_s = (m*r + r*n)*B (with B=32 bits for FP) versus the binarized weight cost m*n*B' (B'=1 bit). Given r ≪ m,n, they show that even if stored in 32-bit, the LRMB adds only a tiny fraction of what the full matrix would, confirming the overhead is minimal. Similar reasoning is applied to the sparse branch: only k (top 0.1%) entries of each weight matrix are kept, so the extra cost is trivial. All these derivations are mathematically correct. There are no complex new proofs in the paper; rather, the authors ensure that each design element is backed by a clear explanation or formula. I did not find any algebraic errors or logical gaps in these derivations. The use of the straight-through estimator (STE) for binarization is referenced (e.g., ReSTE [1]) and the binary convolution operation is formulated via XNOR and bit-count equations – these are standard in binary neural networks and are presented correctly. One minor observation is that the paper doesn’t formally prove why the combination of branches yields optimal retention of information (that would be a very difficult theoretical guarantee). Instead, it relies on intuitive reasoning (e.g., outlier weights are rare but crucial​, so capturing them in SMB is beneficial) which is then validated experimentally. This approach is acceptable for an applied paper. In summary, the theoretical basis of the method is solid and internally consistent.

[1] Wu, Xiao-Ming, et al. "Estimator meets equilibrium perspective: A rectified straight through estimator for binary neural networks training." Proceedings of the IEEE/CVF International Conference on Computer Vision. 2023.

---

> ### Author Rebuttal · Authors · 2025-04-01
>
> > Q2-1：Cite LoRA and KD
>
> A2-1: Thank you for your advice. We will cite LoRA and KD in the revised version.
>
> > Q2-2：One minor claim that could use more direct evidence is the suggestion that BiMaCoSR enables diffusion SR on resource-limited edge devices.
>
> A2-2: In Table 2 in the main paper, our BiMaCoSR takes 1.83 GFlops and 4.98 M parameters, which can be safely deployed and run efficiently on mobile devices, e.g., Snapdragon 8 Gen3 based devices. We will add this example in the revised version.
>
> > Q2-3：Clarify reported speedup vs compression ratios.
>
> A2-3: Currently, the hardware support to binarized model is not well-developed. Therefore, we are not able to provide the actual-device running time. Despite, we report the speedup ratio following the previous research [1,3,4] and calculate the FLOPs needed for the inference process in the same way. Therefore, it is an engineering task to reach the calculated speedup ratio. As for research, we focus more on the balance of performance and speedup ratio.
>
> > Q2-4：Include details on k selection for SMB.
>
> A2-4: We describe the k selection for SML in Sec. 3.4. Specifically, we select the top k values with the highest absolute values of $\mathbf{W}_{\text{BMB}}^{\prime}$ to form the SMB branch. Each element is save with a triple, i.e., (row index, column index, value). During inference, there are efficient algorithms to calculated sparse matrix multiplication.
>
> > Q2-5：Actual Inference Speed
>
> A2-5: Please refer to A2-3.
>
> > Q2-6：Full-Precision Layers: You chose to keep the first and last conv layers in full precision. Did you attempt to binarize those as well, and if so, how badly did it hurt performance? In other words, is this partial precision absolutely required for acceptable results?
>
> A2-6: Experimentally, quantization of the first and the last conv layers hurts the performance significantly with negligible compression effect, as shown in the table below. This quantization scheme is also leveraged in many previous works[1,2,3,4].
>
> | RealSR          | LPIPS↓ | DISTS↓ | FID↓  | CLIP-IQA+↑ | Params  |
> |-----------------|--------|--------|-------|------------|---------|
> | SinSR (FP)      | 0.3635 | 0.2193 | 56.36 | 0.5736     | 118.59M |
> | BiMaCoSR        | 0.3375 | 0.2183 | 86.09 | 0.4800     | 4.98M   |
> | Fully Quantized | 0.3524 | 0.2423 | 91.17 | 0.4617     | 4.92M   |
>
>
>
> [1] Bin Xia, Yulun Zhang, Yitong Wang, Yapeng Tian, Wenming Yang, Radu Timofte, and Luc Van Gool. Basic binary convolution unit for binarized image restoration network. In ICLR, 2022.
>
> [2] Haotong Qin, Mingyuan Zhang, Yifu Ding, Aoyu Li, Zhongang Cai, Ziwei Liu, Fisher Yu, and Xianglong Liu. Bibench: Benchmarking and analyzing network binarization. In ICML, 2023.
>
> [3] Zheng Chen, Haotong Qin, Yong Guo, Xiongfei Su, Xin Yuan, Linghe Kong, and Yulun Zhang. Binarized Diffusion Model for Image Super-Resolution. In NeurIPS, 2024.
>
> [4] BiDM: Pushing the Limit of Quantization for Diffusion Models. In NeurIPS, 2024.

---

### Official Review · Reviewer_XSYS · 2025-03-12

**Overall Recommendation:** 3

**Summary:**

This paper presents BiMaCoSR, a binary one-step diffusion model for efficient real-world image super-resolution (SR), which integrates 1-bit quantization and one-step distillation to address the high computational and memory costs of conventional diffusion models. To mitigate performance degradation caused by extreme compression, a dual-branch compensation mechanism is introduced: the Low-Rank Matrix Branch (LRMB), initialized via top-r singular value decomposition (SVD) to preserve low-frequency information, and the Sparse Matrix Branch (SMB), which maintains high-rank feature representations by absorbing extreme values in feature maps.

These branches synergize with the binarized main branch (BMB) to enable decoupled learning of high- and low-frequency features. The model employs pretrained weights for initialization, with LRMB initialized through SVD decomposition and SMB via sparse value selection.

Evaluations on RealSR and other benchmarks demonstrate superior performance over existing binarization methods in PSNR, SSIM, and LPIPS metrics. Ablation studies confirm the effectiveness of the dual-branch architecture and initialization strategies. This work proves that combining matrix compression techniques with one-step distillation enables efficient deployment of diffusion models on resource-constrained devices while preserving visual quality.

**Claims And Evidence:**

- The claim that "However, applying naive binarization will lead to catastrophic model collapse" is made without citing relevant sources. Moreover, the claim that adding skip connections can resolve this issue also lacks citation. This absence of supporting references undermines the solidity of the motivation presented.

- The low-rank and high-rank of LRMB and SMB don't equate to low- and high-frequency information. For instance, the paper mentions using skip connections (like identity matrices) to access high-frequency information, yet edge detection operators can also convey such information.

- The article claims to first one-step binarized diffusion, but this claim is questionable. Many binarized diffusion works exist in related research, and one-step implementation can be achieved through various sampling methods.

**Essential References Not Discussed:**

No relevant papers were found.

**Experimental Designs Or Analyses:**

The experiment includes effect comparisons, key speed tests, and ablation studies on each module. However, it has flaws: no actual - device running time was designed for speed-related aspects; Table 3 of the ablation study lacks BMB + SMB results, even though SMB's sparse matrix might be low-rank.

**Methods And Evaluation Criteria:**

The method in the paper seems to be a general binarization approach for diffusion models, yet the rationale for validating it on the Sr task is unclear. While the evaluation for the SR task is reasonable.

**Other Comments Or Suggestions:**

No

**Other Strengths And Weaknesses:**

Strengths:
+ The paper's SR performance is excellent, surpassing that of the compared methods in many dimensions.

Weaknesses:
 - The paper's performance is good, but the biggest question is why binarization is tested on SR tasks with an unclear starting point. Testing it on pure generative tasks might yield more insights

**Questions For Authors:**

For binarized SR tasks, its performance is excellent, but I'm puzzled by some unmentioned methods. Inconsistent sampling steps aren't a valid reason for not comparing.
Could this binarization - designed structure be applied to pure generative tasks?

**Relation To Broader Scientific Literature:**

No relevant papers were found.

**Theoretical Claims:**

There are no relevant theoretical proofs in the main text, but rank-related parts are in the supplementary materials. I checked the logic there and found no errors

---

> ### Author Rebuttal · Authors · 2025-04-01
>
> > Q1-1: The claims about naive binarization and skip connection lack citations.
>
> A1-1: It is a common sense that naive binarization leads to model collapse and we provide experiments in table below. Moreover, we will add citations [1], [3], and [4] for naive binarization and [3] and [4] for skip connection in the revised version.
>
> |                          | LPIPS↓ | FID↓   | CLIP-IQA+↑ | Param |
> |--------------------------|--------|--------|------------|-------|
> | BMB (Naïve Binarization) | 0.4141 | 110.15 | 0.4325     | 3.69M |
> | BiMaCoSR                 | 0.3375 | 86.09  | 0.4800     | 4.98M |
>
>
> > Q1-2: The low-rank and high-rank of LRMB and SMB don't equate to low- and high-frequency information.
>
> A1-2: Yes. We clarify that the low-rank and high-rank of LRMB and SMB are not equivalent to low- and high-frequency information. Could you please specify the corresponding line number that you are referring to?
>
> > Q1-3: The article claims to first one-step binarized diffusion, but this claim is questionable.
>
> A1-3: In current research, one-step distillation is the only successful way to achieve one-step diffusion model. Simply changing the sampling methods often lead to model collapse [5,6]. Both one-step distillation and binarization are model compression techniques. The combination of these two techniques does not exist in current research. Therefore, to our best knowledge, BiMaCoSR is the first one-step binarized diffusion model.
>
> > Q1-4: The method in the paper seems to be a general binarization approach for diffusion models, yet the rationale for validating it on the Sr task is unclear.
>
> A1-4: We are working on the binarization of one-step diffusion model on SR task to address the needs of industry. Currently, the industrial companies are in urgent need to compress the SR diffusion models, after which can they deploy these excellent SR models on mobile devices to improve the imaging process. Whereas, general diffusion models have no such application scenario and they are usually deploied on cloud. In order to solve the current practical problem in the industry, we are focusing on the SR task, i.e., binarized one-step diffusion model for SR. Yet, we provide the result of BiMaCoSR compared with BiDM, ReActNet on general diffusion models (DDPM) in the following table. This result also supports our superior performance.
>
> | Methods  | FID     | param  |
> |----------|---------|--------|
> | FP       | 16.8274 | 35.72M |
> | BiDM     | 38.5275 | 4.73M  |
> | ReActNet | 76.8448 | 1.12M  |
> | BiMaCoSR | 37.1792 | 2.02M  |
>
> > Q1-5: No actual-device running time was designed for speed-related aspects.
>
> A1-5: Currently, the hardware support to binarized model is not well-developed. Therefore, we are not able to provide the actual-device running time. Despite, we report the speedup ratio following the previous research [1,4,6] and calculate the FLOPs needed for the inference process in the same way. Therefore, it is an engineering task to reach the calculated speedup ratio. As for research, we focus more on the balance of performance and speedup ratio.
>
> > Q1-6: Table 3 of the ablation study lacks BMB + SMB results, even though SMB's sparse matrix might be low-rank.
>
> A1-6: Thank you for your advice. We provide the result of BMB+SMB below and We will add it in the revised version.
>
> | RealSR    | PSNR    | SSIM   | LPIPS  | CLIP-IQA+ |
> |-----------|---------|--------|--------|-----------|
> | BMB + SMB | 26.3037 | 0.7466 | 0.3901 | 0.4565    |
>
> > Q1-7: ... the biggest question is why binarization is tested on SR tasks with an unclear starting point. Testing it on pure generative tasks might yield more insights.
>
> A1-7: We explain the reason and result in A1-4. Please refer to A1-4 for detailed explanation.
>
> > Q1-8: Could this binarization-designed structure be applied to pure generative tasks?
>
> A1-8: Yes, we apply BiDM and our BiMaCoSR on DDPM on CIFAR-10 and the result is provided in A1-4.
>
>
> [1] Basic binary convolution unit for binarized image restoration network. In ICLR, 2022.
>
> [2] Bibench: Benchmarking and analyzing network binarization. In ICML, 2023.
>
> [3] OneBit: Towards Extremely Low-bit Large Language Models. In NeurIPS, 2024.
>
> [4] Binarized Diffusion Model for Image Super-Resolution. In NeurIPS, 2024.
>
> [5] SinSR: Diffusion-Based Image Super-Resolution in a Single Step. In CVPR, 2024.
>
> [6] BiDM: Pushing the Limit of Quantization for Diffusion Models. In NeurIPS, 2024.

---

### Decision · Program_Chairs · 2025-05-01

**Decision:**

Accept (poster)

**Comment:**

The authors propose a combination of three matrix branches to address the problem of a binary one-step diffusion model for real-world image super-resolution. The three matrix branches SMB, LRMB, and BMB all exist in the literature, but their combination to address the problem is novel. This paper demonstrates promising experimental validation and state-of-the-art performance. In their rebuttals, the authors clearly answer the questions raised by the reviewers and provide detailed explanations for their motivation and additional experimental results. Two of the three reviewers recommend weak accept while the remaining reviewer recommends weak reject. Considering their clear presentation, solid experimental validation, novel combination, and promising performance, I recommend accepting this paper.